# Plant Waste-Based Bioadditive as an Antioxidant Agent and Rheological Modifier of Bitumen

**DOI:** 10.3390/ma17102303

**Published:** 2024-05-13

**Authors:** Valeria Loise, Abraham A. Abe, Michele Porto, Innocenzo Muzzalupo, Luigi Madeo, Maria Francesca Colella, Cesare Oliviero Rossi, Paolino Caputo

**Affiliations:** 1Department of Chemistry and Chemical Technologies, University of Calabria, 87036 Arcavacata di Rende, CS, Italy; valeria.loise@unical.it (V.L.); luigimadeo21@gmail.com (L.M.); mariafrancesca.colella@unical.it (M.F.C.); cesare.oliviero@unical.it (C.O.R.); paolino.caputo@unical.it (P.C.); 2Research Centre for Forestry and Wood, Council for Agricultural Research and Agricultural Economy Analysis, 87036 Rende, CS, Italy; innocenzo.muzzalupo@crea.gov.it

**Keywords:** aged bitumen, antioxidant, light microscopy, olive leaves, rheology

## Abstract

In recent times, circular economy initiatives in addition to the need for sustainable biomaterials have brought about several attempts at the eco-friendly, eco-sustainable and cost-effective production of asphalt pavements. It is an increasingly common practice in the asphalt industry to improve road pavement performance using additives to enhance the physico-chemical properties of bitumen, which performs the role of the binder in the asphalt mix. This paper evaluated the potential of a bio-based additive derived from olive leaf residue as a modifier and antioxidant agent for bitumen. Samples of neat, aged and doped aged bitumen were analyzed. In this study, the two bio-based additives were characterized in terms of phenol, chlorophyll, lignin and cellulose content, which was correlated with the mechanical properties of the tested samples. The mechanical properties of the neat, modified, aged and unaged samples were evaluated via Dynamic Shear Rheology. The bio-based additives proved to be promising and can improve the properties of bitumen binder and the performance of asphalt pavements in general.

## 1. Introduction

### 1.1. Assumptions

Bitumen is viscoelastic material, naturally occurring as such; however, given the scarcity of deposits, it is obtained mainly by crude oil distillation. It is commonly used in road construction due to its role as a binder in asphalt mixes. The susceptibility of asphalt to environmental degradation can limit its service life span. The environmental degradation of asphalt pavements is brought about by oxidative aging which occurs via the oxidation of the bitumen present in the asphalt conglomerate and the evaporation of volatile compounds in the asphalt mix. These aging mechanisms result in an increase in the viscosity, molecular weight and softening point of the bitumen binder and a decrease in its penetration, ductility and stress relaxation capacity [1,2,3,4]. To improve the resistance of asphalt pavements to oxidative aging, research has explored the modification of the bitumen binder in the asphalt mix with various additives, including antioxidants. This paper aims at investigating the effects of olive leaf residue, a bio-based additive derived from the pruning of olive trees, on the properties of bitumen and the potential of this additive to prevent the oxidative aging of asphalt. The idea behind this study is based on the need to enhance the quality and longevity of asphalt pavements in a sustainable and eco-friendly way coupled with the need to find a potential avenue for the use of leaf residues derived from olive tree pruning. In fact, for example, according to Espeso et al. [5], about 1.25 million tons of olive leaf waste are produced every year in Spain. This figure represents only about 50% of world production. This is a considerable amount and it constitutes a key environmental problem. Committing to a circular economy can mitigate problems of this nature, and this can be achieved by finding a valuable use for this sort of waste material.

### 1.2. Olive Leaf Residues

Since ancient times, the olive tree (*Olea europaea* L.) has been a native plant of the Mediterranean basin. This tree produces a large number of leaves throughout its lifespan. In fact, olive tree leaves come either from tree pruning, which is necessary to remove unproductive branches, or from oil production processes in olive mills. Olive leaves are also derived as waste during the production of food-grade olives.

Olive leaf residue is one of the principal by-products of olive farming as it makes up 10% of the entire weight of olive harvest, with a yield of about 7–15 kg per olive tree during the pruning process [6].

Currently, these residues are used in various areas, such as in animal feed, although the residual pesticide content limits the use of olive residues in this area. Olive residues can also be used in biorefineries [7,8,9].

Olive leaf is a lignocellulosic material [10], i.e., a complex structure composed of cellulose (7–15% dw), hemicellulose (6–9% dw) and an aromatic polymer named lignin (14–17% dw). Cellulose is the most abundant sugar monomer of the olive leaf, consisting of a linear polymer of β-1,4 linked glucose units. Cellulose is quite similar to hemicellulose, with the main difference them being the unbranched nature of cellulose. Hemicellulose is a branched polymer composed of sugars which are also found in cellulose, such as xylose and glucose, amongst others. Lignin, however, is mainly stored in the primary cell wall of the plant. Although lignin exhibits certain variations in its chemical composition, depending on the plant of origin, it is still possible to state that it is mainly composed of a polymeric network of phenylpropane units [11]. A thorough knowledge of the qualitative–quantitative composition of olive leaves is essential for designing a more efficient processing plant. In addition, knowledge of their chemical composition is key, as this influences the performance of thermochemical processes for conversion to electrical and thermal energy [12].

The extraction of bioactive compounds, especially phenolic compounds, from olive leaves is one of the most studied research topics [13]. On the other hand, this is not sufficient, and it is important to continue fractionating and recovering other compounds and/or products. Lignocellulosic materials and chlorophylls represent other important bioproducts to be tested in order to achieve the integral use of olive leaves.

The main active phenolic constituent in olive leaves is oleuropein, which can constitute up to 5–10% of dry leaf matter [14]. Oleuropein is the heterotic ester of elenolic acid and hydroxytyrosol and it is the primary molecule responsible for the properties of olive plant products, including their powerful antioxidant activity [15]. Other phenolic compounds which have been found in olive leaves are hydroxytyrosol (the by-product of the degradation of oleuropein), tyrosol, 4-hydroxyphenylacetic acid, verbascoside, ligstroside, and luteolin [14,15,16].

The color of olive leaves is determined by the composition and concentration of pigments, especially chlorophyll. In the dark, these pigments have antioxidant activity [17,18]; however, in the light, they can act as pro-oxidants, leading to the formation of singlet oxygen in the excited state through a reaction with triplet oxygen [19].

In this study, the concentrations of lignocellulosic components, polyphenols and chlorophylls were evaluated in the leaves of the olive trees of Carolea and Tondina.

Olive leaves and their derivatives in general are promising candidates as bitumen additives due to the high levels of antioxidants, polyphenols, and other bioactive compounds that can impart beneficial effects on bitumen. In a previous study, olive plant by-products have been shown to improve the performance of asphalt pavements [20].

### 1.3. Biomass Antioxidant

Although the use of olive tree derivatives as possible antioxidants in bitumen is still an almost unexplored field, many studies have been conducted on the use of antioxidants from biomass. For example, Pahlavan et al. [21] studied the ability to hinder bitumen aging in six phenol-rich bio-oils. Thanks to FTIR, they found that the carbonilic index of the aged samples treated with the oils was lower than that of the net bitumen. According to the authors, this is due to the phenolic compounds present in the tested oils, which play the role of scavengers to quench free radicals formed during the aging process of bitumen. Zhang et al. [20] reached the same conclusions, noting that the effect of the lignin they tested, i.e., phenolic structures containing substituents, was to react with the free radicals that formed during the bitumen aging process, forming stable structures. In this way, the chain reactions generated by free radicals were inhibited. Also Park et al. [22] tested different bio-oils in order to slow down the aging of bitumen. By applying computational models based on DFT, they observed that the bio-oils with the highest phenol content are those that exhibit the greatest antioxidant effect. In a very interesting paper, Pahlavan et al. [23] studied the ability of biomodifiers coming from waste vegetable oil and from wood pellets to delay the aging of rubberized bitumen. They found that the rubberized bitumen containing the biomodifiers had a lower degree of aging than the reference bitumen. In addition, they found that the antiaging property and VOC retention role of a biomodifier largely depends on its molecular composition; in fact, the action of the biomodifier from waste vegetable oil was worse than that from wood pellets, due to the fact that the latter is richer in phenolic compounds. In the 2022, Pahlavan et al. [24] studied the effect that a phenolic compound from plant-based sources has on the crystallization of sulfur in bitumen. They found that the effectiveness of the compounds used was greater the higher their phenol content. According to the authors, phenolic compounds can react with polysulfide radicals stabilizing the polysulfide radicals within the bitumen. Moreover, Zhang et al. [25] tested lignin and lignin with bio-oil as a potential antioxidant for bitumen, and found that lignin exerts a remarkable antiaging effect. Contrariwise, when lignin is used with bio-oil, the antiaging effect worsens. On the other hand, lignin increases viscosity, and the addition of bio-oil decreases it, improving workability. Moreover, Yadykova et al. [26] found that the bio-oil from the fast pyrolysis of clean woody biomass reduces viscosity at high shear stresses and improves the wetting ability of bitumen.

In this study, two additives derived from olive leaf residue were used to modify a 50/70 penetration grade bitumen, and their potential as anti-oxidant agents was investigated via Dynamic Shear Rheology (DSR) and light microscopy. Through laboratory testing and analysis, this study seeks to provide insight into the potential of olive leaf residue as a sustainable alternative for bettering the performance and longevity of asphalt pavements.

## 2. Materials and Methods

### 2.1. Chemicals and Materials

Neat bitumen with a penetration grade of 50/70 supplied by Lo Prete Costruzioni sas was used for the purpose of this research study. Two different bio-based additives labeled as FUR and FUM, derived from the grinding of olive leaves, were added to the bitumen at three different percentages (1.3 and 6% *w*/*w*). The two additives came from the leaves of two different cultivars: the FUR additive from Carolea and the FUM additive from Tondina. A sample of FUM is shown in Figure 1; the FUR sample is the same in appearance and color as the sample shown. The plants for both cultivars were chosen homogeneously in terms of growth, production, phytosanitary protection and age. The plants were grown at the Council for Agricultural Research and Agricultural Economy Analysis in Rende, Cosenza (Italy). The bio-based additives were green powders with an average grain size of approximatively 250 μm.

### 2.2. Sample Preparations

The unaged and aged samples were prepared by modifying bitumen with 3 different additive percentages by weight of the neat bitumen. Aged bitumen was obtained through a Rolling Thin-Film Oven Test (RTFOT) extended to 225 min, according to [27]. As the additives were to be tested as antioxidants, they were added to the bitumen before it underwent aging via RTFOT. The samples were prepared, adding the additives to the hot bitumen (~160 °C) under mechanical stirring (~500 rpm). Samples are stirred for 1 h at 160 °C on a heating plate equipped with temperature control. Furthermore, asphaltenes were extracted from both aged and unaged bitumen samples. Deasphaltenization was conducted with CHCl3 and n-pentane. The same quantities of bitumen, in grams, and chloroform, in milliliters, were manually mixed together in a closed container.

The precipitate, i.e., asphaltenes, was filtered and washed with n-pentane until the solvent became colorless. The asphaltenes were dried in an oven for 3 h.

A list of the studied sample is therefore reported in Table 1, together with the applied IDs.

### 2.3. Experimental Techniques

#### 2.3.1. Rheological Characterization

The samples were tested using a shear stress-controlled rheometer (SR5, Rheometric Scientific, Piscataway, NJ, USA) with parallel plate geometry (gap 2 mm, φ = 25 mm within a temperature range of 25–150 °C) and a Peltier system (±0.1 °C) for temperature regulation (see Figure 2). Dynamic experiments were carried out in triplicate for each sample, within the linear viscoelastic region where the mechanical properties of the material are independent of the amplitude of the applied load and are the only function of the microstructure of the material [28]. Specifically, the samples were analyzed by applying a stress of 100 Pa. The trends of the elastic modulus (G′) and of the viscous modulus (G″) define the response of the material to the application of mechanical stress–strain. The response of the material is temperature-dependent. In fact, when the viscous component becomes predominant, then the material is no longer able to recover from the elastic stress. At this point, the tangent δ, i.e., the ratio between the viscous and elastic moduli, reaches the viscoelastic-to-liquid transition temperature [29,30]. Moreover, from the rheological measurements carried out it was possible to derive the rutting parameter, i.e., the ratio G*sin⁡δ, where $G^{*}$ is given by the sum of the elastic and viscous moduli, a measure of the resistance to rutting of the analyzed sample.

#### 2.3.2. Total Phenol Content

The extraction of polyphenols from the dry powder of olive leaves was performed according to the method described in Difonzo et al. [31] with a few modifications. The olive leaf extract was prepared by an ultrasound-assisted extraction technique using a water/ethanol solution (30:70 (*v*/*v*)) as a solvent. Other parameters, such as time of extraction, temperature and pH were set to the best conditions to favor phenolic content yield as previously reported [32]. Specifically, the extraction was carried out at 25 °C for 20 min at pH = 6.0 using an olive leaf mass-to-solvent volume ratio of 1:5 (*w*/*v*). The liquid extract was centrifuged at 9000 rpm for 10 min, filtered (0.45 µm), dried with a rotavapor, lyophilized and stored at 4 °C in the dark prior to the experiments.

Olive leaves’ phenolic compounds were identified using High-Performance Liquid Chromatography (HPLC) analysis, according to the IOOC method (IOOC, 2009). The extracts (5 mg) were dissolved in methanol (80%, 1 mL). Before HPLC (UV/Vis) analysis, the resulting suspension was filtered using 0.45 μm filters. The HPLC equipment (1100 Series, Agilent, Milan, Italy) was equipped with a UV detector at 280 nm and an integrator, an Inertsil ODS-2 column (5 μm, 15 cm Å~4.6 mm i.d.), a Spherisorb S5 ODS-2 (5 μm, 1 cm Å~4.6 mm i.d., Sigma-Aldrich srl, Milano, Italy) precolumn and an injection volume of 20 μL; the flow rate was 1 mL/min at room temperature. A binary solvent mixture composed of water acidified with 0.2% phosphoric acid (solvent A) and methanol/acetonitrile 50/50 (solvent B) was used as the mobile phase. A linear gradient was run from 96% (A) and 4% (B) to 50% (A) and 50% (B) for 40 min; switched to 40% (A) and 60% (B) for 5 min; and in 15 min, switched to 0% (A) and 100% (B); then, after re-equilibration for 12 min, it returned to the initial composition. The following phenolic standards were used: luteolin, hydroxytyrosol, tyrosol, coumaric acid, ferulic acid, verbascoside, luteolin-7-O-glucoside, rutin, oleuropein, oleuropein aglycon, luteolin-4-O-glucoside, ligstroside, luteolin.

#### 2.3.3. Total Chlorophyll Content

Total chlorophyll (Chl) extraction was carried out according to the methods reported in Muzzalupo et al. [33] with a few modifications. Chl extract was prepared from a 100 mg aliquot of the olive leaf dry powder with 8 mL of acetone/water (4:1, *v*/*v*) twice. Tubes of 50 mL were used to store the liquid phases of the samples. The extraction was repeated thrice until the pellet became colorless. To clear the combined acetone extracts, they were centrifuged at 1500 rpm for 15 min.

Chl content was determined using a spectrophotometer (model Cary 50Bio, Varian, Turin, Italy). A646.8 and A663.2 were determined and used to calculate the total contents of Chl a and b according to the method reported by Lichtenthaler [34]. Three replicates were performed for each replicate, and six measurements were carried out on each sample.

#### 2.3.4. Lignin, Cellulose and Hemicellulose Content

Following the procedure from [35], it was possible to carry out determination of the lignin and structural carbohydrate content. The two-step method consisted of primary hydrolysis by a strong sulfuric acid solution (72 wt%), followed by dilution with water, and secondary high-temperature (125 °C) hydrolysis. According to this procedure, the polymeric carbohydrates were hydrolyzed into soluble monosaccharides, leaving behind a residue which was rich in lignin. This residue was then vacuum-filtered and measured gravimetrically. The carbohydrate fractions of the sample were measured as monomers and liberated in the hydrolysate solution. This process of hydrolysis, when combined with an array of other mathematical tests, provides a quantitative summative compositional analysis of lignocellulosic biomass which ideally accounts for 100% of the original material. Three replicates were performed, and for each replicate, six measurements were carried out on each sample.

#### 2.3.5. Light Microscopy

The asphaltenes from aged and unaged bitumen were placed in a double microscope glass slide (sandwich model), and analyzed using a Leica DMLP polarizing microscope coupled with a Leica DFC280 camera and a CalCTec (Rende, Italy) heating stage. The analysis was carried out at an initial temperature of 120 °C with an increase of 5 °C per minute. In this way, it was possible to determine the melting range of the analyzed asphaltenes [36].

## 3. Results and Discussion

### 3.1. Total Phenol and Chlorophyll Content

As seen in Table 2, FUM has the highest phenolic content, in particular oleuropein, which is the main compound. On the other hand, the amount of each individual phenolic component is lower in FUR. However, it has a greater amount of Chlorophylls, expressed as Chlorophyll (a + b) mg/g of dry weight.

Table 3 shows the content of structural carbohydrates in FUM and FUR. As can be seen, the sample that has the lowest chlorophyll and phenol content (i.e., FUR) is also the one that has the highest lignin and cellulose content. For better interpretation of the data, Table 3 shows the total content of the cell wall.

### 3.2. Rheology and Melting Range

Figure 3 shows the dynamic temperature ramp tests of the investigated samples. As can be seen, FUR has a hardening effect on the neat bitumen, especially at relatively high dosages. In fact, as the temperature increases, the bitumen begins to lose its elasticity and gradually becomes a viscous fluid. The viscoelastic-to-liquid transition temperature is identified as the extreme where the elastic modulus drops and therefore the tangent δ diverges (i.e., Tan δ → ∞). However, the sample modified with 3% FUR has a slightly better hardening effect than the sample modified with 6%, with the high amount of modifier probably causing a saturation effect. On the other hand, FUM (Figure 3b) does not show a substantial improvement in the rheological properties of the bitumen, even at high dosages. In addition, the effect of this additive appears to be independent of the percentage of modification.

Furthermore, in order to evaluate if the action of FUR is due to a mere filler effect, a comparison with the most common inert filler used for road pavements, i.e., CaCO_3_, was made. As can be noted from Figure 3c that the Tan δ trend was almost unaffected by the presence of calcium carbonate.

In order to evaluate the storage stability of modified bitumen, i.e., the phenomenon of powder sedimentation inside the matrix, a tube test was performed [37]. In accordance with EN-13399, the modified bitumen was poured into a cylindrical aluminum tube. The covered tube was vertically placed in an oven at 180 ± 5 °C for 72 ± 1 h. After conditioning, the aluminum tube was placed in a freezer for at least 4 h. After that, the tube was cut into three equal parts. Samples from the top and bottom were heated by removing the metal and analyzed after stirring homogenously. The rheological results of the tube test sample containing the highest dosage of the FUR additive shows that there is no sedimentation effect at all. As can be seen in Figure 4, there is no difference between the rheological profiles of the sample obtained from the top part and the bottom part of the aluminum tube.

By correlating the transition temperatures with the composition of the two FUM and FUR samples, it can be seen that the additive with the highest lignin and cellulose content had a modifying effect on the mechanical properties of the bitumen. This finding is in line with data found in the existing literature, in which both lignin and cellulose are recognized as modifiers of the mechanical properties of bitumen [38,39,40,41].

Furthermore, due to their chemical properties, the two additives were also tested as antioxidants. Aging causes considerable hardening of the binder, shown by the shift of about 20 °C in the sol-gel transition temperature (see Figure 5). As a consequence of the addition of FUM, the bitumen undergoes slowing down of the aging process, i.e., an antioxidant effect. This slowdown is greater in the case of bitumen treated with 1% *w*/*w* of FUM, and decreases with increasing concentrations of the additive (Figure 5b). FUR, on the other hand, does not exhibit antioxidant properties at any concentration tested (Figure 5a).

Contrary to what was previously observed, FUR, which has a lower total content of phenols, does not show any antioxidant activity. It is also important to note that, although FUM contains more chlorophyll than FUR, it manages to counteract the oxidative aging of the bitumen, lowering the transition temperature by about 10 °C compared to aged bitumen. This finding gives the impression that the effect of phenols as natural antioxidants is greater when compared to the effect of chlorophyll, which is a natural pro-oxidant.

The rheological results on the antioxidant effect are confirmed by the light microscopy of the asphaltenes. Table 4 reports the melting range values for the analyzed samples. The asphaltene clusters of the unaged bitumen (LP) begin to melt around 150 °C and are completely liquid at 160 °C. On the other hand, the molecular interactions become stronger with oxidation [30], leading to higher melting range values for asphaltene clusters of aged bitumen (LP aged), which begin to melt at a temperature around 175 °C and are completely liquid at 185 °C. Finally, the asphaltene macromolecules of the bitumen treated with FUM begin to melt at 165 °C and are completely liquid at 170 °C, showing behavior similar to those from unaged bitumen. On the contrary, the sample with FUR shows values close to those recorded for asphaltenes of aged bitumen. Accordingly, FUM chemically interacted with the asphaltenes, protecting them from aging.

Rutting is one of the major stressors of asphalt pavements, caused by high temperature and high traffic loading. According to Singh et al. [42], higher values of the rutting parameter indicate better rutting resistance. Figure 6 shows the trend of the rutting parameter as a function of three different temperatures, chosen in accordance with the literature [43,44,45]. As expected, the rutting parameter decreases with increasing temperature for all analyzed samples. In general, the addition of bio-based additives improves the rut resistance for the unaged sample. The effect of FUR is more pronounced than that of FUM both for unaged and aged samples. Moreover, the rutting parameter increases as the bio-additive content increases. Finally, according to the Superpave specification, the minimum values of the rutting parameter must be >1 kPa and >2.2 kPa for unaged and aged bitumen, respectively [43]. All the analyzed samples reach this threshold at the investigated temperatures. Moreover, the fatigue parameter was also derived, as reported in Figure 7. In general, as expected, the fatigue parameter drops when the temperature increases, both for unaged and aged samples. In addition, it is observed that for unaged samples containing FUM (see Figure 7a), the values of this parameter are extremely close to each other at all three temperatures investigated, whereas for samples containing FUR, the values of the fatigue parameter show a dependence on the percentage of modification. In addition, the difference between the values of the fatigue parameter at the same temperature and concentration is an order of magnitude greater in the case of FUR than FUM. In the case of aged samples (Figure 7b), the parameter values increase with increasing concentration for the FUM additive, while they are extremely similar for the FUR additive.

## 4. Conclusions

This study aimed at unravelling a new additive for bitumen by testing a bio-base additive, bringing about a renewed alternative application to waste from the pruning of olive trees, with a circular economy in view. For this purpose, two additives obtained from the grinding of olive leaves were tested as a modifier and as an antioxidant on bitumen of 50/70 penetration grade. An experimental plan was set up to (1) characterize these two additives through measurements of total chlorophyll content, total phenol content, lignin content, and cellulose and hemicellulose content, and (2) study the mechanical properties of bitumen treated with these additives at different percentages, via rheological measurements and light microscopy. The different techniques were instrumental to the investigation of the exact role that the bioadditives play in improving the performance of bitumen. Rheology was used to evaluate the changes in the mechanical properties between the neat and modified samples. while phenol and chlorophyll content determination gave very valuable insights into the antiaging potentials of the additives in bitumen. Microscopic analysis of the thermal behavior of the asphaltenic fraction of the neat and modified bitumen samples also provided a clear understanding of the effects of the additives on aged and unaged bitumen. In general, the results obtained indicated that the mechanical properties of the modified bitumen samples were enhanced. In more detail, the different compositions, in terms of phenolic and carbohydrate content, are reflected in the different effects that the two additives have on unaged and aged bitumen. In fact, it was shown that the minimum FUM amount tested could protect the bitumen from the aging process. Contrarily, FUR could shift the transition temperature of the base bitumen by about 10 °C. Finally, the authors believe that this research can open new pathways leading to an increase in the reuse of waste materials from agriculture, exploiting the intrinsic properties of plants.

## Figures and Tables

**Figure 1 materials-17-02303-f001:**
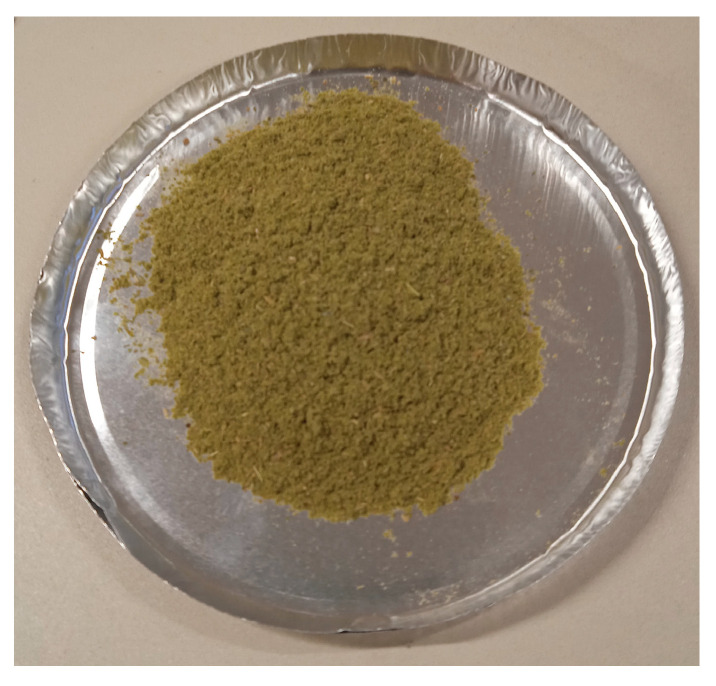
Ground olive leaves.

**Figure 2 materials-17-02303-f002:**
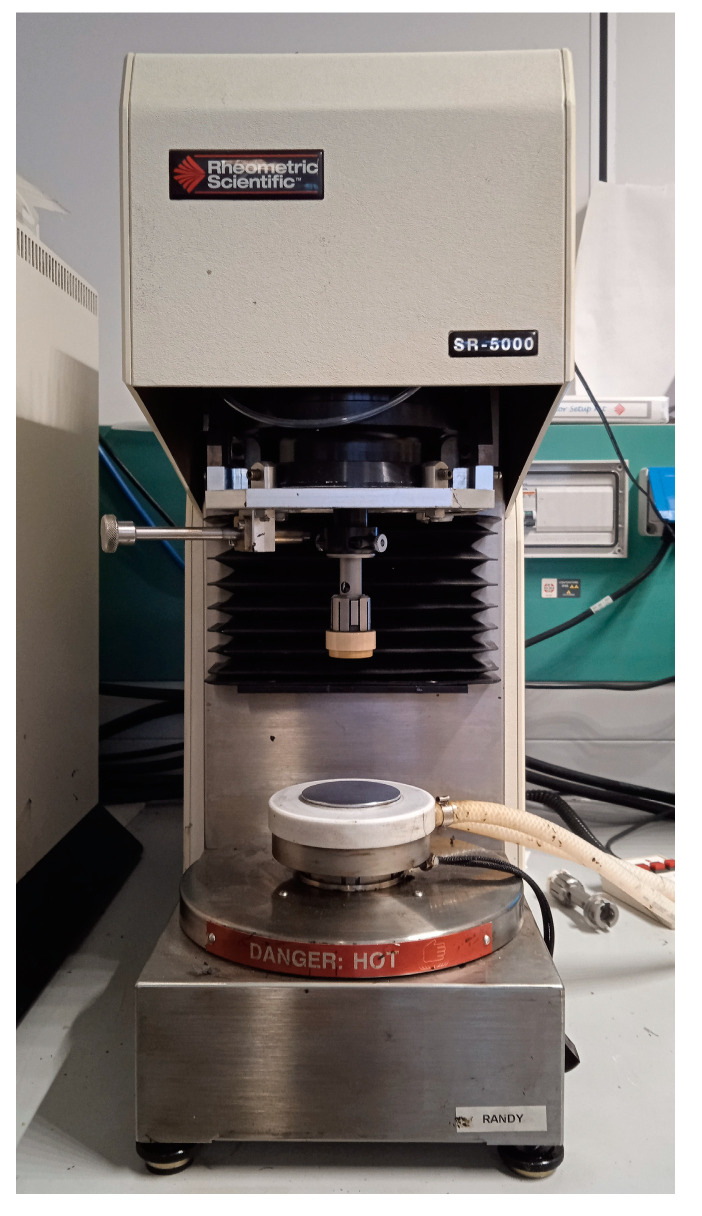
Shear-stress rheometer.

**Figure 3 materials-17-02303-f003:**
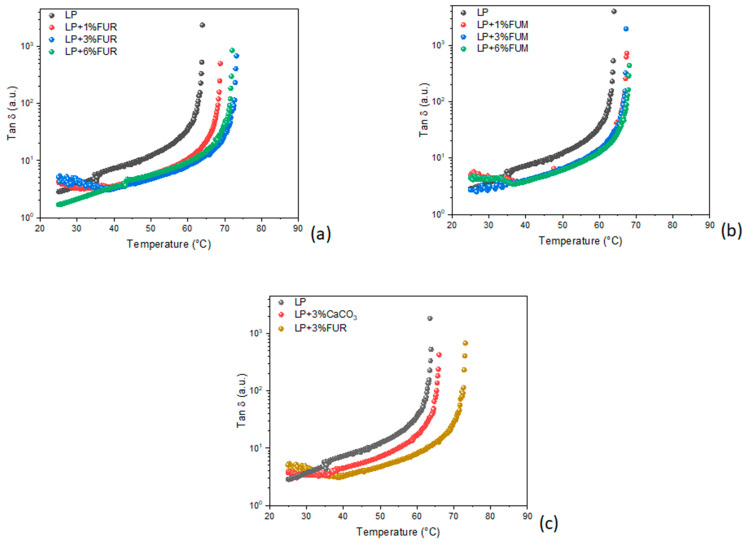
Dynamic temperature ramp tests of the bitumen modified with (**a**) FUR and (**b**) FUM and (**c**) the comparison between modified bitumen with CaCO_3_ and FUR.

**Figure 4 materials-17-02303-f004:**
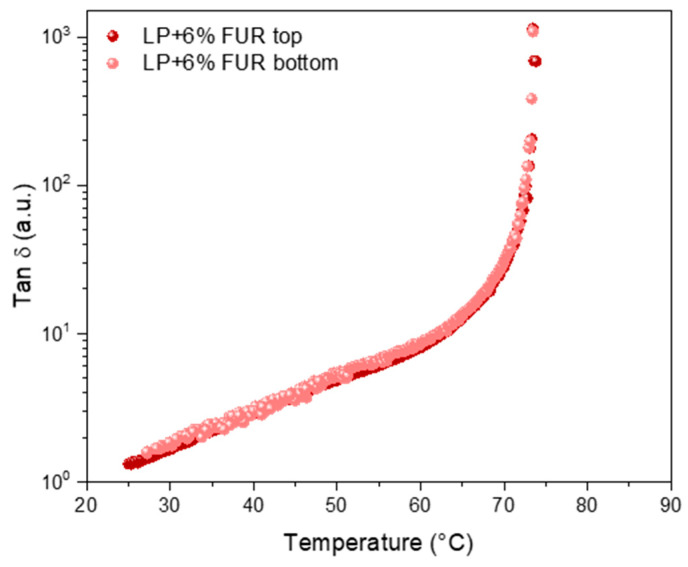
Dynamic temperature ramp tests of the top and bottom tube tests.

**Figure 5 materials-17-02303-f005:**
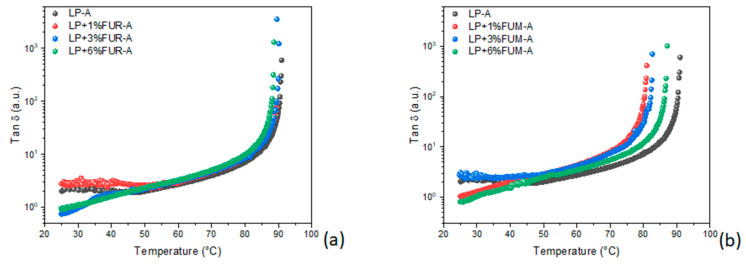
Dynamic temperature ramp tests of the bitumen treated with (**a**) FUR and with (**b**) FUM.

**Figure 6 materials-17-02303-f006:**
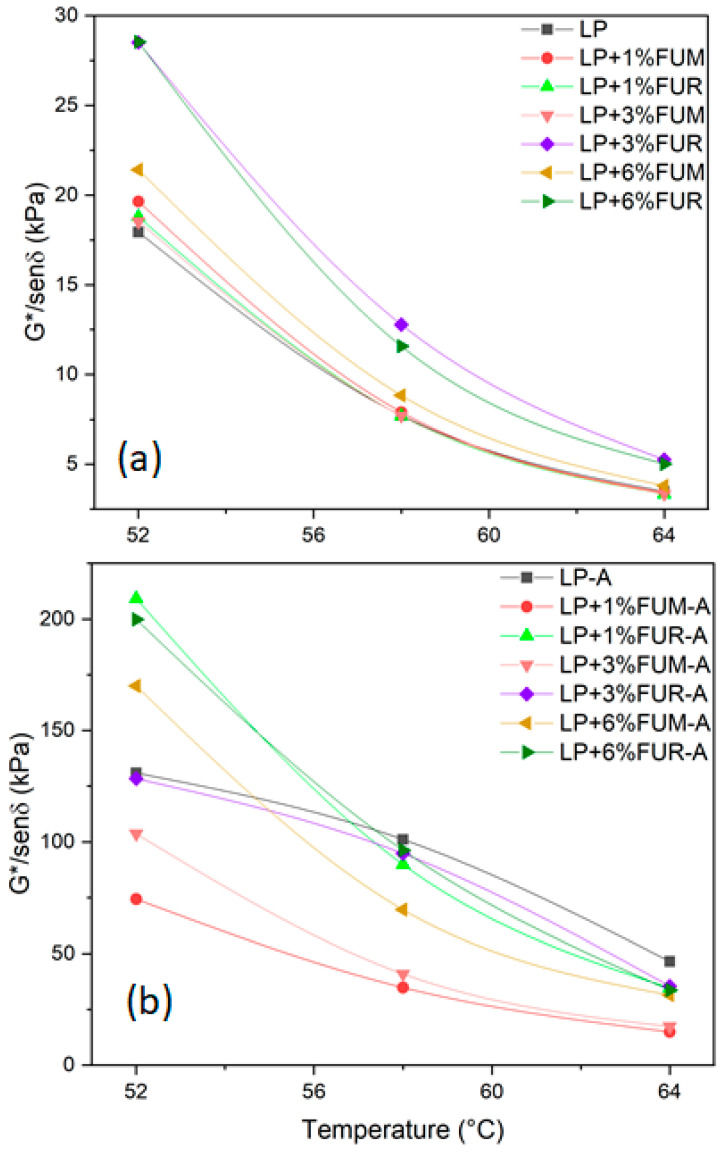
Rutting parameter for the (**a**) unaged samples and (**b**) aged samples.

**Figure 7 materials-17-02303-f007:**
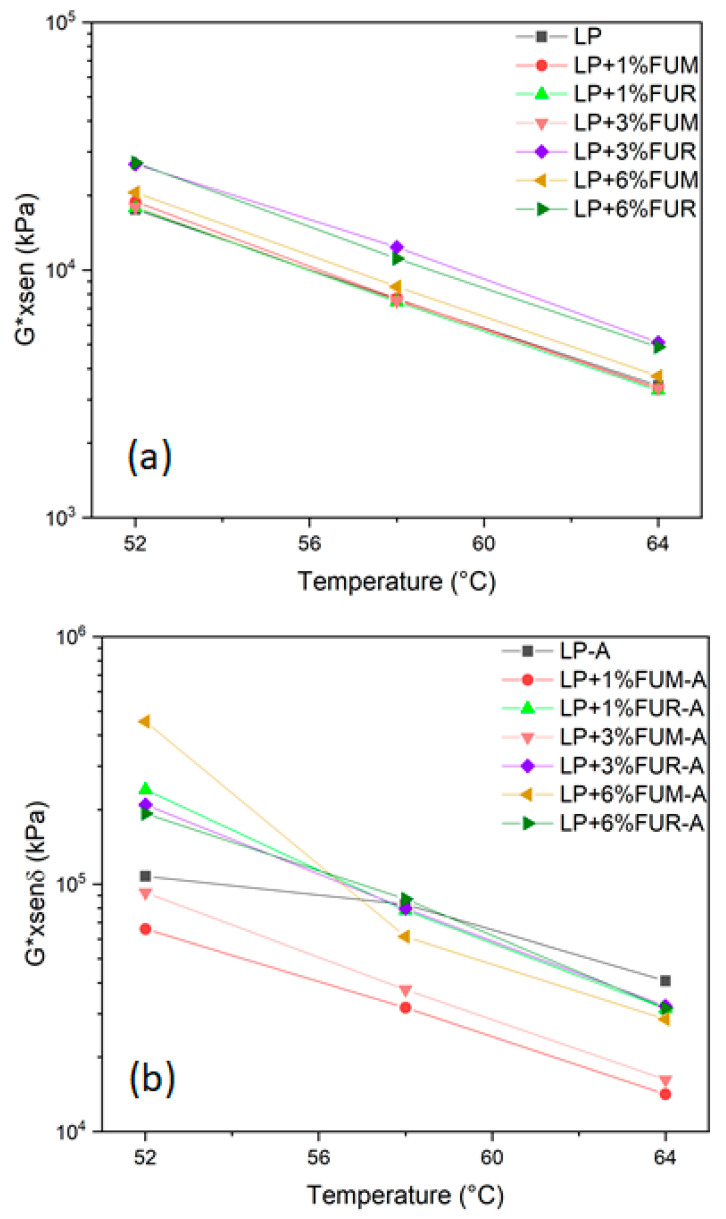
Fatigue parameter for the (**a**) unaged samples and (**b**) aged samples.

**Table 1 materials-17-02303-t001:** List of samples and applied IDs.

Sample	ID	Sample	ID
Bitumen 50/70	LP	Bitumen aged by RTFOT, 225 min	LP-A
Bitumen 50/70 + 1% FUM	LP + 1% FUM	Bitumen aged by 50/70 + 1% FUM RTFOT, 225 min	LP + 1% FUM-A
Bitumen 50/70 + 3% FUM	LP + 3% FUM	Bitumen aged by 50/70 + 3% FUM RTFOT, 225 min	LP + 3% FUM-A
Bitumen 50/70 + 6% FUM	LP + 6% FUM	Bitumen aged by 50/70 + 6% FUM RTFOT, 225 min	LP + 6% FUM-A
Bitumen 50/70 + 1% FUR	LP + 1% FUR	Bitumen aged by 50/70 + 1% FUR RTFOT, 225 min	LP + 1% FUR-A
Bitumen 50/70 + 3% FUR	LP + 3% FUR	Bitumen aged by 50/70 + 3% FUR RTFOT, 225 min	LP + 3% FUR-A
Bitumen 50/70 + 6% FUR	LP + 6% FUR	Bitumen aged by 50/70 + 6% FUR RTFOT, 225 min	LP + 6% FUR-A

**Table 2 materials-17-02303-t002:** Phenolic and chlorophyll composition of FUM and FUR.

	FUM	FUR
**Phenol Totals**	**637.1 ± 8.9 mg/g** **(Dry Weight)**	**544.6 ± 7.7 mg/g** **(Dry Weight)**
Hydroxytyrosol	3.2 ± 0.4	2.1 ± 0.7
Tyrosol	0.3 ±0.1	0.1 ± 0.0
4-Hydroxyphenylacetic acid	17.6 ± 1.9	15.0 ± 1.3
Caffeic acid	1.1 ± 0.4	0.5 ± 0.2
Ferulic acid	1.7 ± 0.6	0.8 ± 0.3
p-Coumaric acid	1.7 ± 0.5	0.8 ± 0.4
Oleuropein	517.8 ± 6.4	450.1 ± 5.5
Verbascoside	26.6 ± 1.9	21.3 ± 1.4
Ligstroside	17.3 ± 1.1	11.4 ± 1.0
Luteolin-7-glucoside	49.8 ± 2.9	42.5 ± 2.6
**Chlorophylls (a + b)**	**2.11 ± 0.9**	**1.83 ± 0.5**

**Table 3 materials-17-02303-t003:** Lignocellulosic and glucose content in the analyzed samples.

	FUM	FUR
	% (Dry Weight)	% (Dry Weight)
Lignin	14.3 ± 0.9	17.2 ± 0.1
Cellulose	7.7 ± 1.1	6.9 ± 0.5
Hemicellulose	7.4 ± 1.5	6.7 ± 0.3
Glucose	6.5 ± 1.2	6.7 ± 0.5

**Table 4 materials-17-02303-t004:** Melting range values for the analyzed asphaltenes.

Sample	Melting Range (±5 °C)
LP	150–160
LP-A	175–185
LP + 1%FUM-A	165–170
LP + 1%FUR-A	170–175

## Data Availability

The raw data supporting the conclusions of this article will be made available by the authors on request.

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
