# Peer review of "Plant Waste-Based Bioadditive as an Antioxidant Agent and Rheological Modifier of Bitumen"

_materials, 2024, doi:10.3390/ma17102303_

Round 1
Reviewer 1 Report
Comments and Suggestions for Authors
The article by Loise V. et al. presents the production of modified bitumen binders using biowaste in the form of olive tree leaves. The authors use two types of biowaste powder from two olive tree varieties as antioxidants and obtain six modified bitumens, which are compared with pure bitumen before and after thermal aging. As a result, the authors show an improvement in the high-temperature stiffness of bitumen under the influence of the modifiers and a smaller increase in stiffness under aging. The authors analyze modifier powders for polyphenol, lignin, and cellulose contents and conclude the effectiveness of modifying additives based on their chemical composition. In principle, the article is interesting and contains new information. Its disadvantage, however, is the limited rheological tests. The authors only provide data on the temperature dependence of the loss tangent. This is extremely low. It is unclear why the authors do not determine the rutting resistance. Moreover, data on resistances to fatigue cracking and low-temperature cracking are lacking. All these data can be obtained with the dynamic shear rheometer used by the authors. In addition, there is a lack of data on the viscosity of modified bitumens. The article contains extremely little rheological data, whereas a lot more data could have been easily obtained with a rheometer.
Other comments are as follows.
- No line numbering, which complicates reviewing.
- “Bitumen is a black, viscous by-product of crude oil distillation”. First of all, bitumen is a natural hydrocarbon. There are different natural bitumens including asphalts, ozokerites, asphaltites, and so on, with different rheological behavior. Since natural bitumens (primarily, asphalts) are low in reserves, they are produced from crude oil and residues of its refining.
- “1.3. Biomass antioxidant”. In this section, the authors describe that the addition of bio-oil improves the ability of bitumen to resist aging by acting as an antioxidant. However, bio-oil also acts as an adhesion enhancer for bitumen binders, increasing their thermodynamic work of adhesion (e.g., 10.1016/j.conbuildmat.2022.129919), which may be pointed out. In the same work, it is also shown that bio-oil gives bitumen non-Newtonian behavior and reduces its high-shear viscosity.
- “to modify a 70/100 penetration grade bitumen” and “Neat bitumen with a penetration grade of 50/70 supplied by Lo Prete Costruzioni”. Contradiction. It is unclear what bitumen the authors used.
- “labelled as FUR and FUM”. Is there any meaning in these acronyms?
- “The bio-based additives are green powders”. It is unclear if there was any drying of the powders. This should be stated. If it was, the temperature and duration should be specified.
- “Furthermore, asphaltenes were extracted from both aged and unaged bitumen samples as was carried out by Oliviero Rossi et al.” The authors should describe the essence of the procedure at least briefly.
- “within the linear viscoelastic region”. The strain amplitude that was used should be specified.
- “Following the procedure according to [34] it was possible to carry out the determination of the lignin and structural carbohydrates content”. The authors should describe the procedure at least briefly.
- “to determine the melting point of the analysed asphaltenes”. This phrase is unclear since asphaltenes are not crystalline compounds. Perhaps the authors mean the transition from a mesophase to an isotropic liquid or from a glass to a flowable liquid.
- “Fig.1 shows the time cure tests”. It is unclear what the authors mean by the phrase "time cure tests".
- “The sol-gel transition temperature”. This is a misleading term since there is no gel formation. More suitable is "solid-liquid transition" or "nominal solid-liquid transition".
- “The sol-gel transition temperature is identified as the extreme where the elastic modulus drops and therefore the tangent δ diverges.” The description is unclear. It is better to show the transition point by an arrow or vertical dashed lines in Figure 1. In addition, you may write that this is the point where tangent δ tends to infinity, i.e., tanδ→∞ (if I understand correctly).
- Table 3 and discussion. The term "melting" is incorrect because asphaltenes are not crystalline.
Comments on the Quality of English LanguageThe English language requires moderate editing.
Author Response
REVIEWER 1
The authors only provide data on the temperature dependence of the loss tangent. This is extremely low. It is unclear why the authors do not determine the rutting resistance. Moreover, data on resistances to fatigue cracking and low-temperature cracking are lacking. All these data can be obtained with the dynamic shear rheometer used by the authors. In addition, there is a lack of data on the viscosity of modified bitumens. The article contains extremely little rheological data, whereas a lot more data could have been easily obtained with a rheometer.
Reply: We thank the reviewer for the comment. We have also included the discussion on rutting and fatigue parameters.
- No line numbering, which complicates reviewing.
Reply: We provided to add the line numbering
- “Bitumen is a black, viscous by-product of crude oil distillation”. First of all, bitumen is a natural hydrocarbon. There are different natural bitumens including asphalts, ozokerites, asphaltites, and so on, with different rheological behavior. Since natural bitumens (primarily, asphalts) are low in reserves, they are produced from crude oil and residues of its refining.
Reply: We thank the reviewer for the comment. We provide to change the sentence as highlighted in the manuscript.
- “1.3. Biomass antioxidant”. In this section, the authors describe that the addition of bio-oil improves the ability of bitumen to resist aging by acting as an antioxidant. However, bio-oil also acts as an adhesion enhancer for bitumen binders, increasing their thermodynamic work of adhesion (e.g., 10.1016/j.conbuildmat.2022.129919), which may be pointed out. In the same work, it is also shown that bio-oil gives bitumen non-Newtonian behavior and reduces its high-shear viscosity.
Reply: we thank the reviewer for the suggestion. We provided to add this other reference.
- “to modify a 70/100 penetration grade bitumen” and “Neat bitumen with a penetration grade of 50/70 supplied by Lo Prete Costruzioni”. Contradiction. It is unclear what bitumen the authors used.
Reply: We are sorry for this error. We modify the text with the correct 50/70 penetration grade bitumen used for this study.
- “labelled as FUR and FUM”. Is there any meaning in these acronyms?
Reply: Yes, these labelled come from Italian “Foglie Ulivo”, i.e. olive leaves. R and M are derived from the place of cultivation of the two different cultivars.
- “The bio-based additives are green powders”. It is unclear if there was any drying of the powders. This should be stated. If it was, the temperature and duration should be specified.
Reply: No further treatment was carried out on the two powders derived from the olive leaves.
- “Furthermore, asphaltenes were extracted from both aged and unaged bitumen samples as was carried out by Oliviero Rossi et al.” The authors should describe the essence of the procedure at least briefly.
Reply: We thank the reviewer for the suggestion. We provide to add a brief explanation.
- “within the linear viscoelastic region”. The strain amplitude that was used should be specified.
Reply: We used the stress-controlled rheometer, all experiments where are performed in the linear viscoelastic region. We determinate this region by the stress sweep test. We applied, as cited in the paper, the stress of 100 Pa, because this value can ensure to be within the linear viscoelastic region.
- “Following the procedure according to [34] it was possible to carry out the determination of the lignin and structural carbohydrates content”. The authors should describe the procedure at least briefly.
Reply: We modified the manuscript according to the reviewer’s comment (highlighted in the manuscript)
- “to determine the melting point of the analysed asphaltenes”. This phrase is unclear since asphaltenes are not crystalline compounds. Perhaps the authors mean the transition from a mesophase to an isotropic liquid or from a glass to a flowable liquid.
Reply: We thank the referee for the right comment. The asphaltenes are both crystalline and amorphous. Indeed, we can talk about range of melting considering the amorphous part of the asphaltenes [DOI https://doi.org/10.1007/s10973-020-09772-y; DOI https://doi.org/10.1080/14680629.2009.9690234] and higher transition order. We modified the text with the more correct “melting range values”
- “Fig.1 shows the time cure tests”. It is unclear what the authors mean by the phrase "time cure tests".
Reply: Time cure test is used as synonym of Dynamic temperature ramp tests. However, we provide to change time cure in dynamic temperature ramp tests.
- “The sol-gel transition temperature”. This is a misleading term since there is no gel formation. More suitable is "solid-liquid transition" or "nominal solid-liquid transition".
Reply: We appreciate the comment of the review. In fact, the term sol-gel make confusion. We think that the more appropriate term could be “viscoelastic-to-liquid”, in order to mark that the system passes from a viscoelastic material (both G’ and G’’ are detectable) to liquid material (where just G’’ is detectable).
- “The sol-gel transition temperature is identified as the extreme where the elastic modulus drops and therefore the tangent δ” The description is unclear. It is better to show the transition point by an arrow or vertical dashed lines in Figure 1. In addition, you may write that this is the point where tangent δ tends to infinity, i.e., tanδ→∞ (if I understand correctly).
Reply: We modify the text according to the reviewer comment, in a previous article we had inserted vertical lines to highlight the transition temperatures (https://doi.org/10.1016/j.colsurfa.2017.01.025). We didn’t mark the transition temperature by an arrow or vertical dashed lines because there are so many Tan δ profiles and this could make confusion to the reader.
- Table 3 and discussion. The term "melting" is incorrect because asphaltenes are not crystalline.
Reply: We modify with the more correct “melting range”
Reviewer 2 Report
Comments and Suggestions for Authors
The paper falls within the scope of the journal, but unfortunately doesn't meet the standard to be considered for publication. Therefore, my conclusion is that the paper should be rejected.
The paper doesn't have enough novelty and contributions to be published in one prestigious journal, as is Materials. The authors have not well elaborated very important field. Generally, the paper has potential, but in its current form can't be considered for publication.
My concern is also related to the similarity index. Most of the journals allow a maximum similarity index of 20% in total and 5% with a single source. This paper has 38% similarity index in total and 8% with single source. Some parts of the paper are only copy-paste from other sources. That isn't allowed.
Author Response
REVIEWER 2
- The article lacks photos showing the additives used in asphalt, modified asphalt samples, and equipment used in the tests.
Reply: We have included some pictures for as requested
- The subpoints (a, b, c) of the figure captions should be at the beginning.
Reply: We provided to change the position of the subpoint
- In Figure 2, the description of the parameter unit is missing on the Y axis.
Reply: Thank you, we are sorry for this forgetfulness, we have inserted the unit
- Please provide the number of samples used for the rheological tests for each case with different percentages of additive.
Reply: We add this information within the text
- Please provide the type of experimental plan used in the investigation (mentioned in the conclusions).
Reply: We proceeded to change the sentence by emphasizing the two steps of the experimental plan and summarizing the techniques used to conduct it.
Reviewer 3 Report
Comments and Suggestions for Authors
The article contains interesting scientific considerations, and in order to increase its scientific value, it is proposed to include the following comments:
1. The article lacks photos showing the additives used in asphalt, modified asphalt samples, and equipment used in the tests.
2. The subpoints (a, b, c) of the figure captions should be at the beginning.
3. In Figure 2, the description of the parameter unit is missing on the Y axis.
4. Please provide the number of samples used for the rheological tests for each case with different percentages of additive.
5. Please provide the type of experimental plan used in the investigation (mentioned in the conclusions).
Author Response
REVIEWER 3
- The article lacks photos showing the additives used in asphalt, modified asphalt samples, and equipment used in the tests.
Reply: We have included some pictures for as requested
- The subpoints (a, b, c) of the figure captions should be at the beginning.
Reply: We provided to change the position of the subpoint
- In Figure 2, the description of the parameter unit is missing on the Y axis.
Reply: Thank you, we are sorry for this forgetfulness, we have inserted the unit
- Please provide the number of samples used for the rheological tests for each case with different percentages of additive.
Reply: We add this information within the text
- Please provide the type of experimental plan used in the investigation (mentioned in the conclusions).
Reply: We proceeded to change the sentence by emphasizing the two steps of the experimental plan and summarizing the techniques used to conduct it.
Round 2
Reviewer 1 Report
Comments and Suggestions for Authors
The authors have improved the article for its publication.
Comments on the Quality of English LanguageThe English language requires moderate editing.
Reviewer 2 Report
Comments and Suggestions for Authors
The authors have made an effort to improve the paper, but my opinion is the same as in the previous round. The similarity index is one big concern.